# Full-Field Vibration Response Estimation from Sparse Multi-Agent Automatic Mobile Sensors Using Formation Control Algorithm

**DOI:** 10.3390/s23187848

**Published:** 2023-09-13

**Authors:** Debasish Jana, Satish Nagarajaiah

**Affiliations:** 1Samueli Civil and Environmental Engineering, University of California, Los Angeles, CA 90095, USA; dj93@ucla.edu; 2Civil and Environmental Engineering, Rice University, Houston, TX 77005, USA; 3Mechnanical Engineering, Rice University, Houston, TX 77005, USA

**Keywords:** full-field sensing, compressive sensing, multi-agent system, mobile sensors, formation control, structural health monitoring

## Abstract

In structural vibration response sensing, mobile sensors offer outstanding benefits as they are not dedicated to a certain structure; they also possess the ability to acquire dense spatial information. Currently, most of the existing literature concerning mobile sensing involves human drivers manually driving through the bridges multiple times. While self-driving automated vehicles could serve for such studies, they might entail substantial costs when applied to structural health monitoring tasks. Therefore, in order to tackle this challenge, we introduce a formation control framework that facilitates automatic multi-agent mobile sensing. Notably, our findings demonstrate that the proposed formation control algorithm can effectively control the behavior of the multi-agent systems for structural response sensing purposes based on user choice. We leverage vibration data collected by these mobile sensors to estimate the full-field vibration response of the structure, utilizing a compressive sensing algorithm in the spatial domain. The task of estimating the full-field response can be represented as a spatiotemporal response matrix completion task, wherein the suite of multi-agent mobile sensors sparsely populates some of the matrix’s elements. Subsequently, we deploy the compressive sensing technique to obtain the dense full-field vibration complete response of the structure and estimate the reconstruction accuracy. Results obtained from two different formations on a simply supported bridge are presented in this paper, and the high level of accuracy in reconstruction underscores the efficacy of our proposed framework. This multi-agent mobile sensing approach showcases the significant potential for automated structural response measurement, directly applicable to health monitoring and resilience assessment objectives.

## 1. Introduction

Bridge health monitoring is essential to ensure public safety, prolong infrastructure lifespan, and mitigate potential risks through continuous assessment of structural integrity and performance. Although fixed sensors placed on the structure are commonly used for vibration-based bridge health monitoring [1], they require ongoing monitoring of sensor health to ensure data reliability [2,3]. Mobile sensing presents an alternative approach, involving the installation of vibration sensors on mobile vehicles or carriers [4]. These mobile units traverse the structure, collecting vibration response data in relation to spatial and temporal variations. Mobile sensing offers distinct benefits compared to traditional fixed sensors, including increased spatial information, scalability, and reduced maintenance costs [5]. The advancements in wireless sensing technologies enable mobile sensor networks to complement wired counterparts, facilitating expanded usage of mobile sensors [6]. Modern smartphones, equipped with motion sensing chips like accelerometers and gyroscopes, enhance the potential for crowd-sourced data collection [7,8,9]. Nevertheless, these mobile sensing techniques often require multiple passes on bridges to record vibration response data. Notably, Matarazzo et al. [10] demonstrated that controlled field experiments and UBER trips on the Golden Gate Bridge enable continuous modal information extraction from smartphone-recorded vibration data. In their study, the researchers traversed the bridge 102 times and utilized 72 UBER trips, capturing acceleration vibration data with smartphones. The collected data were then analyzed to determine the most probable modal frequencies (MPMFs) using the synchro-squeezed wavelet transform [11]. Importantly, multiple vehicles were employed simultaneously for data collection. Additionally, the effectiveness of this mobile sensing approach was verified on the Harvard Bridge by Matarazzo et al. [12]. In a similar vein, Eshkevari et al. [13] conducted an experimental study validating crowd-sourced modal identification using continuous wavelets (CMICW), utilizing a collection of smartphone-mounted sensors on vehicles that moved back and forth across the bridge using a motored pulley system. Therefore, this paper introduces a multi-agent formation control framework to automate the mobile sensing procedure, eliminating the need for manual driving stages, particularly in the context of structural vibration response sensing.

The concept of capturing bridge vibration data through sensors on moving vehicles was first introduced by Yang et al. [14]. Leveraging the pure structural responses recorded by these mobile sensors, system properties have been deduced using input–output balance [15]. Over the past decade, extensive studies have explored mobile sensing through diverse avenues, encompassing analytical and numerical analyses [16,17,18,19], laboratory-scale experiments [20,21,22,23], and real-life scenarios [24,25]. The literature predominantly emphasizes bridge modal identification via mobile sensing. Oshima et al. [26] detected mode shape-based support damage by mapping mobile sensor data to fixed sensor data. Mode shape-based bridge damage detection was achieved by Malekjafarian and O’Brien [27] using the short time-frequency domain decomposition (STFDD) method. High-resolution mode shapes were obtained via laser vibrometer and accelerometers mounted on vehicles as proposed by O’Brien and Malekjafarian [28]. Additional signal processing techniques like Short-Time Fast Fourier Transform (STFFT) [29], Empirical mode decomposition (EMD) [30,31], and Hilbert transform [32] are employed for estimating mode shapes from data collected by mobile sensors. Matarazzo et al. [33,34] introduced the “structural identification using expectation maximization (STRIDE)” method for mode shape identification from mobile sensors. Eshkevari et al. [35,36,37] formulated mobile sensing data as a sparse matrix with missing values. They employed alternating least-square (ALS) for matrix completion, followed by principal component analysis (PCA) and structured optimization analysis (SOA) for modal identification. Matrix completion approaches have gained traction in recent years for health monitoring due to their data-driven nature, applicable to both fixed sensors [38] and mobile sensors [19,36,37]. Yang and Nagarajaiah [38] utilized nuclear norm minimization for matrix completion, and a comprehensive overview of such methods is presented in Nagarajaiah and Yang [39].

Throughout the aforementioned research, instances involving multiple mobile sensors for structural sensing or system identification typically involve independent manual control of each vehicle by humans. In certain cases, trains or vehicles with multiple trailers [26] have been employed, attaching sensors to each axle. The evolution of self-driving cars [40] presents the potential to streamline and enhance the mobile sensing process for structural health monitoring (SHM). Multiple self-driving cars could be useful for this purpose; however, autonomous vehicles are optimized for individual operations, often proving expensive for structural vibration response measurement work. Thus, an alternative approach is imperative to automate the mobile sensing procedure without relying on costly self-driving vehicles.

The primary objective of this paper is to automate the mobile sensing process instead of manually driving the vehicles or deploying fully self-driving cars. As a novel contribution, this paper introduces a formation control-based framework to collect the bridge vibration response data through multi-agent systems. The vibration measurement sensors installed on the multi-agent can capture structural responses at the corresponding position of their movement on the system, which can create a sparse space–time response matrix. In recent work, we proposed full-field structural vibration response estimation from a limited number of fixed sensors using data-driven [41] and physics-based [42] approaches. In this study, we deploy this concept to acquire full-field structural vibration responses but utilizing a limited number of automatic multi-agent mobile sensors.

In this paper, we first introduce a framework designed for acquiring full-field responses using mobile sensors and underscore its significance in Structural Health Monitoring (SHM) for bridges. Within this proposed framework, two primary components take center stage: the formation control strategy and compressive sensing. We provide a brief overview of these fundamental concepts. The formation control strategy streamlines the process by harnessing mobile sensors, while the compressive sensing algorithm is employed to estimate full-field responses with a limited sensor setup. This becomes especially pertinent as we leverage data from a network of multi-agent sensors to achieve this objective. Subsequently, we validate the effectiveness of this proposed framework through a numerical study involving a simply supported bridge. We explore two distinct scenarios for formation control. Additionally, we outline practical implementation recommendations that should be taken into consideration. Finally, we delve into the results, draw conclusions, and outline potential avenues for future research.

## 2. Proposed Framework and Its Significance in Bridge SHM

In this section, we initially outline the overarching process for vibration response sensing for bridge health monitoring. Subsequently, we delineate where our proposed automatic formation control-based mobile sensing framework fits in the domain of bridge SHM. Following this, we introduce the details of the formation control-based mobile sensing framework.

The overall sensing pipeline for the bridge structural health monitoring is illustrated in Figure 1. Generally, such sensing can be performed using either fixed or moving/mobile sensors. For fixed sensing cases, sensors are strategically placed at specific locations on the bridge. This approach facilitates the measurement of vibration responses at those particular bridge locations. By analyzing these responses, the health of the bridge can be monitored by estimating system parameters, detecting the presence of damage, and deriving a full-field response from a limited array of sensors. This full-field response can then be utilized for tasks such as damage localization, full-state estimation, or control purposes. An alternative approach to obtaining similar information involves mobile sensing. In this setup, sensors are mounted on vehicles, which can be operated manually or autonomously. Given that the aim of this paper revolves around achieving a fully automated sensing process with minimal human intervention, we refrain from addressing manual driving. Also, achieving coordination through manual driving is complex. In the context of automated driving, two possibilities emerge: (a) autonomous vehicles like Tesla or Waymo, and (b) multi-agent systems wherein vehicles interact with each other. Self-driving cars entail substantial costs, and using multiple self-driving cars for sensing purposes might prove economically unfeasible, as a single sensor might not suffice for bridge health monitoring. Thus, this paper concentrates on multi-agent systems to perform such tasks, as they offer a cost-effective and automated approach to the sensing process.

Nevertheless, within all mobile sensing strategies—whether manual driving or automatic driving—the responses measured by the sensor installed within the vehicle comprise bridge responses, superimposed with road roughness and vehicle dynamics [43], attributed to vehicle–bridge interaction [44,45]. When the recorded sensor data are amalgamated with the vibration from other sources mentioned above, conventional SHM methods cannot be directly applied as they are designed to work with pure structural vibration responses [46]. To circumvent this problem, the existing literature addresses primary approaches: (1) controlling the sensing conditions like vehicle speed and road roughness intensity so that the recorded response mainly contains the bridge vibration [14,27], (2) modeling the vehicle–bridge interaction in a closed form to eliminate the uncertainties due to vibration sources other than the bridge itself [47,48], (3) to use blind source separation (BSS) techniques to extract the different sources of the recorded response [35]. The BSS technique is capable of estimating pure bridge vibration response.

In this paper, we focus on the multi-agent-system-based sensing framework and assume that the recorded bridge vibration response has undergone prior processing to eliminate the unintended road roughness and vehicle–bridge-interaction-related motion to work with pure structural vibration. This is a valid assumption as we explore only numerical scenarios in this study.

In the formation-control-based framework, the multi-agent system is designed to traverse the bridge in a user-defined formation. At a given instantaneous position of all the mobile agents, they collect the vertical vibration response of the bridge. The ultimate goal of this paper is to estimate full-field vibration response time history. In terms of matrix terminology, the objective is to complete the spatiotemporal response matrix. The mobile agent sparsely populates some of the elements of this spatiotemporal response matrix. To better elucidate the matrix completion process, drawing a comparison with fixed sensors could provide enhanced clarity.

The distinction between fixed and mobile sensing for a simply supported bridge is depicted in Figure 2. In scenarios involving fixed sensors, the spatiotemporal response matrix is populated along a particular column based on the sensor’s position, as illustrated in Figure 2a. Conversely, with mobile sensors, the spatiotemporal response matrix is populated diagonally in accordance with the movement and instantaneous positions of the vehicles, as shown in Figure 2b. The slope of these diagonals depends on the vehicle speeds. Now, considering one particular time instance marked by the red arrow in Figure 2b, only four elements are occupied (as four sensors are considered for demonstration purposes). The spatial compressive sensing algorithm’s task is to estimate other values from the measured ones. Consequently, executing this procedure in real time for all rows leads to a complete spatiotemporal matrix, essentially constituting the full-field vibration response.

The authors recently published studies to complete the spatiotemporal matrix using the compressive sensing algorithm for fixed sensor cases [41,42], addressing the problem in Figure 2a. In the present paper, we intend to employ the same algorithm to accomplish spatiotemporal matrix completion for mobile sensor cases, e.g., addressing the problem in Figure 2b. In essence, the core proposition of this paper revolves around introducing a formation control algorithm to efficiently and autonomously populate the elements of the spatiotemporal matrix using a multi-agent system. Subsequently, the capability of compressive sensing is leveraged to complete the spatiotemporal response matrix utilizing response data collected through the multi-agent system, thus estimating the full-field vibration response of the structure.

## 3. Formation Control of Multi-Agent System Formulation

In this section, we introduce the formation control algorithm intended for automated sensing within the multi-agent system. We adopt a graphical model to depict interactions among these multi-agent systems, a prevalent approach in the state-of-the-art study. This framework often utilizes a graph, wherein agents represent graph nodes, while the graph edges symbolize communication and sensing exchanges between these agents, as emphasized by Godsil and Royle [49].

Graph G is defined as (V,E); here, V denotes the set of nodes or vertices of graph and E⊆V×V symbolizes the edges. Here, it is assumed that there are no self-edges, viz., (i,i)∉E for any i∈V, which is a valid assumption. The neighbor set of node i∈V is defined as $N_{i}:=\left\{j\in V:\left(i,j\right)\in E\right\}$. The graph edges are weighted by $w_{ij}$ which are associated with (i,j) for i,j∈V; here, $w_{ij}>0$ if (i,j)∈E and $w_{ij}=0$ otherwise. The Laplacian Matrix $L=\left[l_{ij}\right]\in R^{\left|V|\times |V\right|}$ of G=(V,E) is defined as
(1)$$l_{ij}=\left\{\begin{array}{cc} \sum_{k\in N_{i}}w_{ik}, & \mathrm{if}\;i=j\\ -w_{ij}, & \mathrm{if}\;i\neq j \end{array}\right..$$

The majority of dynamic models for automated mobile sensors adhere to the double-integrator dynamic models, given that the control feedback can be linearized in this form, as outlined in Ren and Beard [50]. In the context of graph G, a collection of *n* agents, each modeled as a double-integrator system, residing within an *m*-dimensional space, can be succinctly expressed:(2)$$\left\{\begin{array}{c} {\dot{p}}_{i}=v_{i},\\ {\dot{v}}_{i}=u_{i}, \end{array}\right.\;i=1,\cdots ,n.$$

Here, $p_{i}\in R^{m}$, $v_{i}\in R^{m}$, and $u_{i}\in R^{m}$ indicate the position, velocity, and control input of agent *i* in relation to the global coordinate system. The agents have the capacity to sense the relative positions and velocities of neighboring agents within the global coordinate system. The overarching aim of the agents is to maneuver in a manner that controls a formation shape, where the designated desired position $p^{*}\in R^{mn}$ and desired velocity $v^{*}\in R^{mn}$ are pre-defined.

A formation comprises agents that move to acquire and maintain a specific geometric configuration based on the relative positions of neighboring agents. A formation can be denoted by specifying the desired position $p_{i}^{*}\left(t\right)$ and the desired velocity $v_{i}^{*}\left(t\right)$ for all i=1,⋯,n at any given time instant t≥0. The primary goal is to formulate a control strategy $u_{i}$ in a manner such that,
(3)$$\left\{\begin{array}{c} p_{i}\left(t\right)⟶p_{i}^{*}\left(t\right),\\ v_{i}\left(t\right)⟶v_{i}^{*}\left(t\right), \end{array}\right.\;i=1,\cdots ,n.$$Here, the symbol, “⟶”, indicates the desirability, e.g., the actual position and actual velocity of agent *i* at time instant *t* which are $p_{i}\left(t\right)$ and $v_{i}\left(t\right)$, but desired to be $p_{i}^{*}\left(t\right)$ and $v_{i}^{*}\left(t\right)$.

If the agents can precisely measure their positions and velocities in the global coordinate system in which $p_{i}^{*}\left(t\right)$ and $v_{i}^{*}\left(t\right)$ are specified, then the Equation (Equation 3) could be solved in a straightforward manner using classical control. In this case, the controller follows the subsequent equation:(4)$$u_{i}\left(t\right)=g_{p}\left(p_{i}^{*}\left(t\right)-p_{i}\left(t\right)\right)+g_{v}\left(v_{i}^{*}\left(t\right)-v_{i}\left(t\right)\right);\;i,j=1,\cdots ,n;\;g_{p},g_{v}>0.$$

In this context, parameters $g_{p}$ and $g_{v}$ function as scaling factors linked to the position and velocity components of the control force. The formulation of formation control presented in Equation (Equation 4) corresponds to the concept of position-based formation control, a framework previously explored in works such as Ren and Beard [51] and Oh et al. [52]. However, within this position-based control scheme, every agent must possess sophisticated sensors capable of precisely measuring position and velocity with respect to global coordinates. Implementing this control strategy could prove challenging due to the associated financial costs tied to the requirement for advanced sensors, especially only for structural health monitoring purposes. Nevertheless, if agents are restricted to sensing their neighboring agents’ relative positions and velocities solely, the goal outlined in Equation (Equation 4) becomes notably more intricate to attain. A more lenient objective emerges, revolving around maintaining relative positions and velocities amongst the agents. This approach is termed displacement-based control [52]. In this context, agents actively regulate their neighboring counterparts to realize the intended formation, with most agents operating without knowledge of the global coordinate system. Consequently, a less rigid objective is to devise a control law $u_{i}$ such that
(5)$$\left\{\begin{array}{c} p_{i}\left(t\right)-p_{j}\left(t\right)⟶p_{i}^{*}\left(t\right)-p_{j}^{*}\left(t\right),\\ v_{i}\left(t\right)-v_{j}\left(t\right)⟶v_{i}^{*}\left(t\right)-v_{j}^{*}\left(t\right), \end{array}\right.\;i,j=1,\cdots ,n.$$

To satisfy the objective in Equation (Equation 5), the control law can be written as
(6)$$\begin{array}{cc} u_{i}\left(t\right) & =g_{p}\sum_{j\in N_{i}}\left(p_{j}\left(t\right)-p_{i}\left(t\right)-p_{j}^{*}\left(t\right)+p_{i}^{*}\left(t\right)\right)\\ \; & +g_{v}\sum_{j\in N_{i}}\left(v_{j}\left(t\right)-v_{i}\left(t\right)-v_{j}^{*}\left(t\right)+v_{i}^{*}\left(t\right)\right);\;g_{p},g_{v}>0;\;i=1,\cdots ,n. \end{array}$$

Considering the formation moves in a constant velocity, ${\dot{p}}^{*}=v^{*}$ and ${\dot{v}}^{*}=0$. We assume
$$p=\left\{\begin{array}{c} p_{1}\\ \vdots \\ p_{n} \end{array}\right\};\;v=\left\{\begin{array}{c} v_{1}\\ \vdots \\ v_{n} \end{array}\right\};\;p^{*}=\left\{\begin{array}{c} p_{1}^{*}\\ \vdots \\ p_{n}^{*} \end{array}\right\};\;v^{*}=\left\{\begin{array}{c} v_{1}^{*}\\ \vdots \\ v_{n}^{*} \end{array}\right\}.$$

From Equation (Equation 2), the system dynamics are
(7)$$\left\{\begin{array}{c} \dot{p}\\ \dot{v} \end{array}\right\}=\left[\begin{array}{cc} 0_{mn} & I_{mn}\\ 0_{mn} & 0_{mn} \end{array}\right]\left\{\begin{array}{c} p\\ v \end{array}\right\}+\left[\begin{array}{c} 0_{mn}\\ I_{mn} \end{array}\right]u.$$

Equation (Equation 6) can be rewritten as
(8)$$u_{i}=g_{p}[|N_{i}|\left(p_{i}^{*}-p_{i}\right)-\sum_{j\in N_{i}}\left(p_{j}^{*}-p_{j}\right)]+g_{v}[|N_{i}|\left(v_{i}^{*}-v_{i}\right)-\sum_{j\in N_{i}}\left(v_{j}^{*}-v_{j}\right)].$$Here, $\left|N_{i}\right|$ denotes the cardinality of neighbor set $N_{i}$ (total number of neighbors of agent *i*). Equations (Equation 7) and (Equation 8) can be simplified to
(9)$$u=\left[\begin{array}{cc} g_{p}\left(L_{n}⊗I_{m}\right) & g_{v}\left(L_{n}⊗I_{m}\right) \end{array}\right]\left\{\begin{array}{c} p^{*}-p\\ v^{*}-v \end{array}\right\}.$$Here, ⊗ denotes the Kronecker product. Defining the error signals as
(10)$$\left\{\begin{array}{c} e_{p}=p^{*}-p\\ e_{v}=v^{*}-v \end{array}\right.,$$$e_{p}$ and $e_{v}$ are the differences between the desired and actual amplitude of agent position and velocity, respectively. With ${\dot{p}}^{*}=v^{*}$ and ${\dot{v}}^{*}=0$, the error dynamics can be evolved from Equations (Equation 2) and (Equation 10) as
(11)$$\left\{\begin{array}{c} {\dot{e}}_{p}={\dot{p}}^{*}-\dot{p}=v^{*}-v=e_{v}\\ {\dot{e}}_{v}={\dot{v}}^{*}-\dot{v}=-\dot{v}=-u \end{array}\right..$$

From Equations (Equation 9) and (Equation 11),
(12)$${\dot{e}}_{v}=-u=\left[\begin{array}{cc} -g_{p}\left(L_{n}⊗I_{m}\right) & -g_{v}\left(L_{n}⊗I_{m}\right) \end{array}\right]\left\{\begin{array}{c} e_{p}\\ e_{v} \end{array}\right\}.$$

From Equations (Equation 11) and (Equation 12), error dynamics expression [52] is
(13)$$\left\{\begin{array}{c} {\dot{e}}_{p}\\ {\dot{e}}_{v} \end{array}\right\}=\left[\begin{array}{cc} 0_{mn} & I_{mn}\\ -g_{p}\left(L_{n}⊗I_{m}\right) & -g_{v}\left(L_{n}⊗I_{m}\right) \end{array}\right]\left\{\begin{array}{c} e_{p}\\ e_{v} \end{array}\right\}.$$

The system represented in Equation (Equation 13) reaches consensus if and only if $G_{n}$ is connected (if there is at least one edge from one node to any other node of the graph, $G_{n}$, is said to be connected). In consensus,
(14)$$\left\{\begin{array}{c} ||e_{p_{i}}\left(t\right)-e_{p_{j}}\left(t\right)||\to 0,\\ ||e_{v_{i}}\left(t\right)-e_{v_{j}}\left(t\right)||\to 0, \end{array}\right.\;t\to \infty .$$

The rate at which convergence or consensus is achieved hinges on the values of the constants $g_{p}$ and $g_{v}$, intrinsic properties of the controller. In the context of Structural Health Monitoring (SHM) applications involving carriers, these constants, $g_{p}$ and $g_{v}$, are reliant upon the vehicle controller responsible for steering the system. During convergence or consensus processes, Equation (Equation 14) guarantees compliance with Equation (Equation 5). With the control law established as per Equation (Equation 6), the dynamics of the closed-loop system are characterized by
(15)$$\left\{\begin{array}{c} \dot{p}\\ \dot{v} \end{array}\right\}=\left[\begin{array}{cc} 0_{mn} & I_{mn}\\ -g_{p}\left(L_{n}⊗I_{m}\right) & -g_{v}\left(L_{n}⊗I_{m}\right) \end{array}\right]\left\{\begin{array}{c} p\\ v \end{array}\right\}+\left[\begin{array}{cc} 0_{mn} & 0_{mn}\\ g_{p}\left(L_{n}⊗I_{m}\right) & g_{v}\left(L_{n}⊗I_{m}\right) \end{array}\right]\left\{\begin{array}{c} p^{*}\\ v^{*} \end{array}\right\}.$$

In the scenario where agents are limited to sensing only the relative positions and velocities of neighboring agents, they would not fulfill Equation (Equation 3), implying an inability to attain predetermined absolute positions within the global coordinate system. To address this, a minimum subset of agents, usually just one, must possess the capability to sense absolute positions. This particular agent plays the role of a leader within the multi-agent system, and by employing a leader–follower methodology, formation consensus can be achieved. Consequently, the control law is modified as follows:(16)$$\begin{array}{c} u_{i}\left(t\right)=g_{p}\sum_{j\in N_{i}}\left(p_{j}\left(t\right)-p_{i}\left(t\right)-p_{j}^{*}\left(t\right)+p_{i}^{*}\left(t\right)\right)\\ +g_{v}\sum_{j\in N_{i}}\left(v_{j}\left(t\right)-v_{i}\left(t\right)-v_{j}^{*}\left(t\right)+v_{i}^{*}\left(t\right)\right)\\ +h_{i}g_{p}\left(p_{l}^{*}-p_{l}\right)+h_{i}g_{v}\left(v_{l}^{*}-v_{l}\right), \end{array}$$
where $p_{l}$ and $v_{l}$ denote the position and velocity of the leader, respectively, and
(17)$$h_{i}=\left\{\begin{array}{cc} 1 & \mathrm{if}\;i=l\\ 0 & \mathrm{otherwise} \end{array}\right..$$

Defining the matrix, $H=\mathrm{diag}(h_{1},\cdots ,h_{n})$, the error dynamic is given by
(18)$$\left\{\begin{array}{c} {\dot{e}}_{p}\\ {\dot{e}}_{v} \end{array}\right\}=\left[\begin{array}{cc} 0_{mn} & I_{mn}\\ -g_{p}\left[\left(L_{n}+H_{n}\right)⊗I_{m}\right] & -g_{v}\left[\left(L_{n}+H_{n}\right)⊗I_{m}\right] \end{array}\right]\left\{\begin{array}{c} e_{p}\\ e_{v} \end{array}\right\},$$
and similarly, the final closed-loop system dynamics is expressed as
(19)$$\begin{array}{cc} \left\{\begin{array}{c} \dot{p}\\ \dot{v} \end{array}\right\} & =\left[\begin{array}{cc} 0_{mn} & I_{mn}\\ -g_{p}\left[\left(L_{n}+H_{n}\right)⊗I_{m}\right] & -g_{v}\left[\left(L_{n}+H_{n}\right)⊗I_{m}\right] \end{array}\right]\left\{\begin{array}{c} p\\ v \end{array}\right\}\\ & +\left[\begin{array}{cc} 0_{mn} & 0_{mn}\\ g_{p}\left[\left(L_{n}+H_{n}\right)⊗I_{m}\right] & g_{v}\left[\left(L_{n}+H_{n}\right)⊗I_{m}\right] \end{array}\right]\left\{\begin{array}{c} p^{*}\\ v^{*} \end{array}\right\}. \end{array}$$

In brief, Figure 3 presents a flowchart illustrating the formation control strategy. This strategy comprises two distinct loops: the inner loop focuses on controlling the individual agent dynamics, as described in Equation (Equation 2), while the outer loop manages the overall formation, adhering to the less stringent objective outlined in Equation (Equation 5).

The performance evaluation of the proposed formation control algorithm can be based on the disparity between the desired and actual positions of the agents [53]. Therefore, for any given agent *i* and time instance *t*, the error is quantified as $\Delta_{i}\left(t\right)=\left|p_{i}^{*}\left(t\right)-p_{i}\left(t\right)\right|$. Over the entire data collection duration denoted as *Q*, the position error for each agent is computed as $\Delta_{i}=\frac{1}{Q}\sum_{i=1}^{Q}\Delta_{i}\left(t\right)$. Ultimately, the collective formation error for all *n* agents is expressed as $\Delta =\frac{1}{n}\sum_{i=1}^{n}\Delta_{i}$. It is important to note that since the primary goal of the multi-agent system is to populate the elements of the spatiotemporal response matrix, the formation error is exclusively contingent on the agents’ positions and not their velocities.

## 4. Brief Overview of Full-Field Response Estimation from a Limited Number of Sensors

Using the Formation control algorithm proposed in Section 3, the elements of the spatiotemporal response matrix are sparsely filled as shown in Figure 2b. In order to obtain the full-field response, the compressive sensing framework proposed by the authors [41,42] is used in this paper. Hence, in this section, we briefly discuss the procedure for the sake of completeness.

### 4.1. Compressive-Sensing-Based Full Signal Reconstruction from Few Measurements

The concept of Compressive Sensing [54] is briefly discussed in this section. A signal $y\in R^{m}$ is sparse, if
(20)$$y=\mathrm{Dx}=\sum_{j=1}^{n}x_{j}d_{j}=\sum_{j\in S}x_{j}d_{j}.$$

In this context, $D\in R^{m\times n}$ signifies the orthonormal basis matrix, with $d_{j}$ representing the *j*th column of D. Typically, the basis matrix is treated as overcomplete, i.e., m<n. The majority of coefficients of $x_{j}$ are zero in Equation (Equation 20). This characteristic results in signal sparsity, which can be expressed as $S=\{j|x_{j}\neq 0\}$. The level of sparsity is represented by $s=\left|S\right|={\left||x|\right|}_{0}$, thus indicating that $x\in R^{n}$ represents a sparse vector.

The Compressive Sensing (CS) technique is capable of estimating $y\in R^{m}$ from the noisy measured vector $z\in R^{p}$, where p<<m.
(21)z=Θy+e=ΘDx+e=Φx+e,whereΦ=ΘD,
where $\Theta \in R^{p\times m}$ constitutes the measurement matrix. The term e signifies the error or noise constrained within the bound ${\left||e|\right|}_{2}\leq \epsilon$. Consequently, the estimation of basis coefficients is attainable by solving the following convex optimization problem:(22)$$\hat{x}=\underset{||\Phi x-{z||}_{2}\leq \epsilon }{arg min}{\left||x|\right|}_{1},$$
where the $\ell_{2}$ norm is represented by $\left|\right|\cdot {\left|\right|}_{2}$. The formulation given in Equation (Equation 22) can be expressed within an optimization framework known as LASSO [55], as follows:(23)$$\mathrm{minimize}\;||\Phi x-{z||}_{2}+{\lambda ||x||}_{1}.$$Here, λ represents the regularization parameter. The interior point method [56] is employed to derive the sparse solution x from Equation (Equation 23), subsequently enabling the recovery of the complete signal y using Equation (Equation 20).

As proposed by Amini et al. [57], the determination of the minimum sampling points required for accurate signal reconstruction relies on the basis matrix D. This estimation can be achieved by applying techniques like Singular Value Decomposition (SVD) and Normalized Power Index (NPI), as follows:(24)$$D=U\Sigma V;\;{\mathrm{NPI}}_{p}=\frac{\sum_{i=1}^{p}\sigma_{i}^{2}}{\sum_{i=1}^{m}\sigma_{i}^{2}},$$
where $D\in R^{m\times n}$, $U\in R^{m\times m}$, $V\in R^{n\times n}$, and $\Sigma \in R^{m\times n}$ with m<n. Diagonal values of Σ represent the singular values, and $\sigma_{i}$ represents the *i*th singular value. The minimum sensor number for accurate signal reconstruction is the smallest integer value of *p* for which NPI→1.

We use the concept of compressive sensing in the spatial domain for every time instant to obtain the full field response from the response time histories of sparse sensors. However, the knowledge of basis or Dictionary matrix D in Equation (Equation 20) is still required. If no model knowledge is available, then Dictionary learning [58] can be used to obtain D from the training signals [41]. On the contrary, if the intrinsic physics of the system is known, then physics-informed dictionaries could be easily obtained [42]. Both of these methods are briefly discussed.

### 4.2. Learning the Basis Functions Using Dictionary Learning

Dictionary learning [58] designs matrix $D\in R^{m\times n}$ to attain a good sparse representation y≈Dx for a set of signals $y\in R^{m}$ based on training samples. The sparse vectors, $x\in R^{n}$, consist of few nonzero coefficients. To construct the dictionary, D, matrix $Y\in R^{m\times N}$ can be formulated, where columns correspond to training signals and *N* represents the number of training signals. Assembling this matrix Y involves arranging individual y signals in a stack. Consequently, the optimization problem inherent to Dictionary learning can be expressed as follows:(25)$$\begin{array}{cc} \min_{D,X}\; & ||Y-{\mathrm{DX}||}_{F}^{2}\\ \mathrm{such}\;\mathrm{that},\; & \left|\right|x_{\ell }{\left|\right|}_{0}\leq s,\;\ell =1:N,\\ \; & \left|\right|d_{j}\left|\right|=1,\;j=1:n. \end{array}$$Here, X corresponds to the matrix of sparse representations, while $\left|\right|\cdot {\left|\right|}_{F}$ denotes the Frobenius norm. Upon resolving the optimization problem defined in Equation (Equation 25), each column within the matrix D serves as a basis function for the signal set Y.

Directly obtaining D and X from Y is difficult as Y=DX; hence, it is subdivided into two smaller optimization problems: (a) Sparse Coding and (b) Dictionary Updating. Basically, in Dictionary learning, the objective is to obtain **D** and **X** from the training signals **Y**. The typical approach for solving such challenges is alternating minimization, which involves the following steps: (1) During the sparse coding phase, **D** remains fixed while **X** is estimated, and (2) in the Dictionary updating stage, **X** is held constant while **D** is estimated. This iterative process continues until a convergence point is reached.

### 4.3. Obtaining the Basis Functions from Physics-Based Knowledge

One method alternative to Dictionary learning for estimating basis matrix D is obtained from the closed-form solution of the inherent differential equation of the continuous system. One example of a simple beam is presented in this section. The equation of motion governing an Euler–Bernoulli beam subject to a distributed transverse force can be denoted using the formulation given by Rao [59]:(26)$$\frac{\partial^{2}}{\partial x^{2}}\left[EI\left(x\right)\frac{\partial^{2}w\left(x,t\right)}{\partial x^{2}}\right]+\rho A\left(x\right)\frac{\partial^{2}w\left(x,t\right)}{\partial t^{2}}=f\left(x,t\right),$$
where w(x,t) signifies the transverse displacement response of the beam, while f(x,t) represents the applied forcing function. Here, *E* denotes Young’s modulus, ρ represents density, and I(x) and A(x) stand for the moment of inertia and cross-sectional area at position *x* from one end of the beam, respectively. In the case of uniform beams, it is reasonable to assume that the transverse displacement response can be expressed as a linear combination of the beam’s normal modes utilizing a separation of variables approach. For a simply supported beam of length *L*, the deflection equation is expressed as
(27)$$w\left(x,t\right)=\sum_{i=1}^{\infty }W_{i}\left(x\right)\eta_{i}\left(t\right)=\sum_{i=1}^{\infty }C_{i}\sin\left(\beta_{i}x\right)\eta_{i}\left(t\right)=\sum_{i=1}^{\infty }C_{i}\sin\frac{i\pi x}{L}\eta_{i}\left(t\right)=\sum_{i=1}^{\infty }{\tilde{C}}_{i}\sin\frac{i\pi x}{L}.$$

Here, the *i*th mode is characterized by the mode shape $W_{i}\left(x\right)$ in generalized coordinates. The response time history of the *i*th mode is denoted as $\eta_{i}\left(t\right)$. The spatial parameter β is connected to the natural frequency ω through relation $\omega =\beta^{2}\sqrt{\frac{EI}{\rho A}}$. The constant $C_{i}$ represents the amplitude associated with the *i*th mode, dependent on the applied forcing function. Consequently, the basis matrix D can be formulated as
(28)$$D=\left[\begin{array}{ccc} \sin(\beta_{1}x_{1}) & \cdots & \sin(\beta_{n}x_{1})\\ \sin(\beta_{1}x_{2}) & \cdots & \sin(\beta_{n}x_{2})\\ \vdots & \vdots & \vdots \\ \sin(\beta_{1}x_{m}) & \cdots & \sin(\beta_{n}x_{m}) \end{array}\right].$$

When considering *p* random measurements across the beam’s length at a specific time instance, the representation can be expressed [60] as follows: $z_{j}=\sum_{q=1}^{n}C_{q}^{*}\sin\left(\beta_{q}x_{j}\right);j=1,2,\cdots ,p$. This can be compactly represented in matrix form as z=ΘDx=Φx. In this context, $x={\left[C_{1}^{*},C_{2}^{*},\cdots ,C_{n}^{*}\right]}^{T}$, and it is expected that the sparse solution should exhibit non-zero values for $C_{q}^{*}$ if $C_{q}^{*}\approx {\tilde{C}}_{i}$. It is crucial to note that the sparse solution x differs from spatial locations $x_{i}$.

In summary, Figure 4 illustrates the compressive sensing framework used to estimate the full-field vibration response, employing a network of multi-agent sensors.

## 5. Numerical Analysis and Result

In this section, we showcase the practicality of the proposed formation control framework, as introduced in Section 3, for accomplishing essential formations within multi-agent systems. Initially, we demonstrate the feasibility of achieving user-defined formations from the initial condition of automated mobile sensors. The term “intended formation” refers to the prescribed motion of a collection of mobile sensors, while “initial condition” pertains to the initial position and velocity of this group of mobile sensors. Subsequently, we illustrate the process of estimating the full-field response by employing recorded responses from the array of mobile sensors, using the method outlined in Section 4. In this context, all mobile sensors are treated as point sensors, and their mass is negligibly small in comparison to the total mass of the bridge, which is a realistic assumption. For this numerical analysis, a simply supported bridge is considered as the structure of interest.

### 5.1. System Description

Bridge decks could be modeled as simply supported beams. In this study, a simply supported steel beam [41] of 50 m in length, 1 m in height, and 0.5 m in width is considered. The beam’s Young’s modulus is $E=2.1\times 10^{11}$ Pa, while its density is ρ=7860 kg/m${}^{3}$. Consequently, the first four natural frequencies of the beam are computed as 0.94, 3.75, 8.44, and 15 Hz. For this scenario, a 1% Rayleigh damping is taken into consideration. The total count of virtual and dense sensing points is set at 4999, resulting in a spatial separation of 0.01 m between the virtual sensing points. The primary aim of this section involves the estimation of vibration time histories for the dense virtual points (4999 in total) using the vibration data collected from a limited number of mobile sensors. In this example, we acquire the system’s basis functions through Dictionary learning [41]. In practical scenarios, the training data required for Dictionary Learning can be acquired through various means, including the utilization of cameras [61,62] or alternative full-field sensing techniques such as Digital Image Correlation [63,64]. Utilizing this data-driven basis matrix, compressive sensing is applied across the entire time series to acquire the time histories of all virtual sensing points. Subsequently, the obtained dense time histories need to be compared with against the results of finite element formulation to assess the accuracy of reconstruction. To facilitate this evaluation, a relative error metric $\epsilon_{i}$ [41] is considered as follows:(29)$$\epsilon_{i}=\frac{\left|\right|R_{\mathrm{Exact},i}-R_{\mathrm{Estimated},i}{\left|\right|}_{2}^{2}}{\frac{1}{m}\sum_{i=1}^{m}\left|\right|R_{\mathrm{Exact},i}{\left|\right|}_{2}^{2}}\times 100;\;i=\mathrm{sensor}\;\mathrm{index}.$$Here, *m* is the number of virtual sensing points, and $\left|\right|\cdot {\left|\right|}_{2}$ indicates the two-norm. $R_{\mathrm{Estimated},i}$ and $R_{\mathrm{Exact},i}$ symbolize the estimated and exact (FEM) responses of the *i*th virtual sensing point, respectively. Both $R_{\mathrm{Exact},i}$ and $R_{\mathrm{Estimated},i}$ have dimensions of $n_{t}\times 1$, where $n_{t}$ corresponds to the number of time samples. The overall average error E [41] is expressed as follows:(30)$$E=\frac{1}{n_{s}}\sum_{i=1}^{n_{s}}\epsilon_{i},$$
where E signifies the average error of all the relative errors $\epsilon_{i}$ of independent virtual responses, and hence E is invariant to the number of virtual sensors.

### 5.2. Different Types of Formation Control and the Corresponding Reconstruction Result

In this section, we examine two specific configurations termed Formation-1 and Formation-2, with the objective of assessing the capability of the proposed formation control to emulate user-defined formations. In the context of Formation-1, the fleet of mobile sensors traverses the entire bridge back and forth, capturing vibration responses. In contrast, Formation-2 involves the mobile sensors moving back and forth within localized sections of the bridge. Subsequent sections comprehensively delve into the details of these formations. Each individual mobile sensor captures the acceleration response of the bridge’s vibration data, a practical choice owing to the convenience of installing accelerometers on vehicles.

#### 5.2.1. Formation-1

This formation involves a total of ten mobile sensors. Half of these sensors (R${}_{1}$ to R${}_{5}$) commence from the left segment of the bridge, while the remaining half (R${}_{6}$ to R${}_{10}$) initiate their movements from the bridge’s right segment, as depicted in Figure 5a. Subsequently, R${}_{1}$ to R${}_{5}$ advance towards the right extremity of the bridge until the leading vehicle detects proximity to the bridge’s end. Likewise, R${}_{6}$ to R${}_{10}$ move towards the bridge’s left end. Through synchronized back-and-forth motion, all these mobile sensors record the vibration response data from the bridge. The number of laps conducted entirely depend upon the user’s data collection duration. Since the mobile sensors move in opposing directions, creating crossover instances, practical feasibility warrants of two lanes, as depicted in Figure 5.

As the direction of movement of agents R${}_{1}$ to R${}_{5}$ and R${}_{6}$ to R${}_{10}$ are opposite, crossing occurs between these two sets of agents in the middle region of the bridge. During these occurrences, two agents simultaneously record identical bridge vibration response readings. Consequently, in such scenarios, the elements of the spatiotemporal response matrix are determined as the average measurements derived from these two mobile sensors. The graph connection topology between all the mobile agents is visually depicted in Figure 6. Here, graph G is considered as an undirected and unweighted graph, where (j,i)∈E only if $w_{ij}=w_{ji}=1\;\forall ;\left(i,j\right)\in E$ for the sake of simplicity. This study assumes that graph G has no switching topologies (multi-agent connection topologies remain unchanged over time). The exploration of more intricate agent connection topologies is reserved for future endeavors.

Analyzing the graph connection topology in Figure 6, it is evident that the entire graph is disconnected. R${}_{1}$ to R${}_{5}$ are interlinked, while R6 to R10 are connected independently. In these scenarios, mobile sensors are connected in a manner where each vehicle can only sense the relative position and velocity of its nearest neighboring mobile sensors. To maintain a global sense of position, at least one mobile sensor, the leader, must measure its position relative to the global coordinates (Equation (Equation 16)). For the connected graph of R${}_{1}$ to R${}_{5}$, R${}_{1}$ is designated as the leader; similarly, R${}_{6}$ serves as the leader for R${}_{6}$ to R${}_{10}$. Importantly, any vehicle within R${}_{1}$ to R${}_{5}$ or R${}_{6}$ to R${}_{10}$ could be assigned as the leader. The reference velocity amplitude $v^{*}$ is set to 1 m/s for all vehicles, though this choice is user-dependent. However, this can result in the mobile agents leaving the bridge after traversing from one end to the other. The multi-agent system must move back and forth to capture longer vibration data, as depicted in Figure 5. This movement can be achieved by applying the same control strategy discussed in Section 3 at different time windows. The formation error for Formation-1 is calculated as 0.98 m; it arises due to the minimal interconnectivity among the agents. To achieve a more precise formation, it is possible to increase communication among the agents, but this would come at the cost of higher computational demands, particularly concerning wireless communication between the agents. It is important to note, however, that this formation error does not affect the estimation of the full-field response matrix since compressive sensing is employed for this purpose. Compressive sensing is well-suited for reconstructing the entire signal from randomly selected samples. In this context, compressive sensing effectively generates the complete spatial profile of the full field from the randomly positioned multi-agent vehicles at a given time instance. With a time sampling frequency of 100 Hz, this demonstration involves the automated mobile sensors collecting data for 100 s. Consequently, the spatiotemporal matrix assumes dimensions of 10,000 × 4999 (with 4999 spatial points as discussed in Section 5.1). Data from the automated mobile sensors populate some of the spatiotemporal response matrix elements, while the rest are filled using the compressive sensing algorithm outlined in Section 4. The spatiotemporal response matrix elements filled by the automatic multi-agent mobile sensors are depicted for four-time instances (at t = 5, 10, 15, 20 s) in Figure 7.

Figure 8a showcases all the sparsely populated entries in the spatiotemporal response matrix for the entire 100 s using Formation-1. Given this limited amount of data, matrix completion or full-field sensor data reconstruction is performed for each time instance through the compressive sensing technique outlined in Section 4. The spatial profile of relative errors $\epsilon_{i}$ for the estimated full-field response, as defined in Equation (Equation 30), is computed and presented in Figure 8b. The computed average error E amounts to 1.18%. Notably, relatively large $\epsilon_{i}$ values are observed at locations around 7.02 m and 44.63 m from the left end, corresponding to relative errors of 6.94% and 1.55%, respectively, as depicted in Figure 8b. Worth mentioning is the higher error values near the bridge’s ends (0–10 m and 40–50 m from the left end), likely due to fewer sensors being present as vehicles cross each other, particularly during instances like t = 12 s, 37 s, and so on. Given that the vehicles are concentrated in the middle portion nearly half the time, this phenomenon contributes to prominent errors at the ends and negligible errors in the middle. Investigating optimal vehicle movement strategies to minimize reconstruction errors across the entire beam could be a scope of future research.

Reconstructed responses for Location 1 are displayed in Figure 9 for two distinct time segments: 50–55 s and 60–65 s. Figure 8a shows that during the period from 50 to 55 s, the instantaneous location of the automated mobile sensors covers the entirety of the bridge, resulting in the reconstructed time history to be exact to the actual true response. Conversely, in the time span of 60–65 s (Figure 8b), as the automated mobile sensors are intercepting near the middle of the bridge, they become concentrated within that specific region. This concentration leads to discernible disparities between the actual and reconstructed time histories.

#### 5.2.2. Formation-2

The reconstruction error of Formation-1 is higher near the bridge ends than in the middle section, which is attributed to the multi-agent system’s crossing near the bridge midpoint. As the final objective is to achieve highly accurate full-field response estimation, Formation-2 is designed to circumvent situations involving “crossing”. In Formation-2, the mobile sensors execute back-and-forth movements within a confined spatial range, as depicted in Figure 10. In this case, we study with only six mobile sensors, as obtained through the formula for the optimal number of sensors required for accurate full-field response estimation as provided in Equation (Equation 24). The graph connection topology among all the automated multi-agent mobile sensors is depicted in Figure 11.

Figure 11 shows that the mobile agents are only connected with their neighboring agents, enabling them to sense their relative velocity and relative displacement with respect to the nearest neighboring vehicles. Consequently, the presence of a leader is necessary to determine global position coordination and ensure proper formation maintenance. For Formation-2, the role of the leader is assumed by the agent R${}_{1}$. The reference velocity amplitude is randomly chosen and kept constant as [1, 1.1, 1.4, 1.6, 1.2, 1.1] m/s for all vehicles throughout the process. Similar to Formation-1, the control strategy is applied in different time windows to achieve the back-and-forth movement of the automatic multi-agent mobile sensors. The formation error of Formation-2 is calculated as 0.54 m and it is important to emphasize that this formation error does not impact the estimation of the full-field response, as explained in Section 5.2.1. The sparsely filled entries contributed by the automated mobile sensors in the spatiotemporal response matrix of dimensions 10,000 × 4999 are depicted in Figure 12 for four distinct time instances.

Figure 13a shows the sparsely populated entries of the spatiotemporal response matrix for the total 100 s, when the multi-agent system follows Formation-2. Utilizing the compressive sensing technique, the full-field sensor time history can be derived for each time instance using these sparse entries. The spatial distribution of relative reconstruction error for Formation-2 is depicted in Figure 13b. The average relative error is computed as 0.36%. Notably, Location 1 (18.9 m from the left end) and Location 2 (45.6 m from the left end) exhibit relatively higher relative error values of 0.62% and 1.63%, respectively. The corresponding estimated time histories for these locations are depicted in Figure 14, revealing that the reconstructed response time histories are comparable to the actual response time histories.

We attempted to compare the compressive-sensing-based spatiotemporal matrix completion approach with other state-of-the-art algorithms currently available. Nevertheless, employing matrix completion techniques based on Singular Value Decomposition (SVD) methods [65] and the OptSpace method [66] proved unfeasible due to their failure to converge within an acceptable tolerance limit, even with a large number of iterations. This outcome can be attributed to the dimensions of sparse matrices, which are 10,000 × 4999, containing only 10,000 × 6 populated values, resulting in a mere 0.12% of filled entries. Consequently, estimating the remaining unknown values without additional information proved exceedingly difficult. In contrast, our proposed approach applies compressive sensing to each row of the spatiotemporal matrix independently, leveraging knowledge of the underlying basis function obtained either through dictionary learning or a physics-based approach. This approach rendered full-field response estimation feasible, distinguishing it from the other methods.

Please note that different types of formations yield varying levels of reconstruction accuracy. As demonstrated in the aforementioned examples, Formation-2 exhibits superior accuracy in estimating full-field vibration responses compared to Formation-1. The primary objective of this paper is to enable user-driven control of the multi-agent system. It is evident from the agent formation control that the proposed algorithm effectively adheres to user-defined formations. Generally, the compressive sensing-based framework yields improved reconstruction accuracy when mobile sensors can continuously span the entire beam. This is in contrast to Formation-1, which yields inferior results due to instances where all mobile sensors cluster in the central portion of the structure. Identifying the optimal formation remains a potential area for future research, which can be motivated from optimal input [67,68,69] and sensor location [70] for structural system identification literature.

### 5.3. Achieving Formation-1 from Any Initial Condition

While Figure 5a and Figure 10a illustrate the initial starting positions of Formation-1 and Formation-2, respectively, a significant advantage of formation control is its capability to achieve any formation from varying initial positions and velocities of the automated multi-agent mobile sensors using the controller outlined in Equation (Equation 19). Notably, the multi-agent system requires a certain amount of time to converge to the desired formation from any given position or velocity. Throughout this study, $g_{p}$ and $g_{v}$ in Equation (Equation 19) are consistently set to one, controlling the convergence rate towards the formation. A practical example is depicted in Figure 15, wherein the initial positions of the mobile sensors are set at 25 m from the left end (bridge midpoint) with zero initial velocity. Utilizing this starting condition, Formation-1 is achieved. Figure 15d illustrates the formation’s attainment in approximately 26 s. Consequently, for better accuracy while using the full-field response estimation framework, data collected beyond 26 s are appropriate.

The requirements for consensus for the undirected graphs are as follows [50]:(a)Every agent must be connected with at least one other agent; otherwise, achieving consensus becomes unattainable.(b)The time it takes for all agents to reach a consensus from an arbitrary starting point, known as the convergence time is dependent on $g_{p}$, $g_{v}$, and the second eigenvalue of the Laplacian matrix (L in Section 3). Moreover, this convergence time is inversely proportional to the magnitude of the second eigenvalue, which is influenced by the graph connection weights. In essence, increasing the strength of graph connections or weights results in quicker convergence for achieving consensus.(c)The convergence time of consensus is also influenced by graph connectivity. In this study, the multi-agent system is considered to be connected with only neighboring agents. For instance, if we consider the second eigenvalue of the Laplacian matrix as $\lambda_{2}$, considering a total of *n* agents, there can be 2n2 potential graphs, considering the isomorphic graphs as different graphs. Amidst these diverse graph sets, there are instances where the second eigenvalue of the Laplacian matrix ${\hat{\lambda }}_{2}$ exceeds $\lambda_{2}$. Such graphs with a higher second eigenvalue converges faster toward consensus than the neighboring connection graph, as demonstrated in this paper. However, more connections among agents would be attributed to the cost. Therefore, in the pursuit of simplicity and cost effectiveness, we opted to investigate the most straightforward scenarios, such as multi-agent connection with only neighboring agents.

The presented formation control strategy holds potential for modern structural health monitoring through the crowd-sensing of bridge vibration data. It offers the ability to automate the process of collecting vibration data by coordinating the movement of vehicles. The data gathered from smartphones installed in these mobile vehicles can be harnessed to characterize the modal properties of bridge structures under real-world circumstances, which is essential for condition assessment and damage detection frameworks. We discuss the advantages of full-field sensing from a limited number of fixed sensors in detail in our previous studies [41,42]; this paper performs a similar task, i.e., full-field sensing, but with a limited number of automatic mobile sensors.

## 6. Recommendation for Practical Implementation

Practical implementation recommendations are crucial for the successful execution of full-field vibration response estimation tasks based on multi-agent formation control. Several factors must be taken into account when designing experiments for this purpose.

**Sensor arrangement**: To successfully achieve the formation control, every agent should be equipped with the IMU (Inertial Measurement Unit) and Wireless Communication Modules. IMUs combine accelerometers and gyroscopes to measure an agent’s linear acceleration and angular velocity. They are essential for estimating an agent’s orientation and motion dynamics. In this paper, the movement of agent is one-directional; hence, only linear accelerometer is sufficient. For the 2D and 3D formation control problem, IMUs would be necessary. Wireless communication modules (e.g., Wi-Fi, Bluetooth) facilitate communication and coordination among agents as the formation control often requires agents to exchange information with one another. As the proposed formation control requires a “leader” agent, a GPS (Global Positioning System) device should be mounted on the “leader” agent. GPS sensors provide accurate global position information, including latitude, longitude, and altitude, which are used to obtain absolute position estimates of agents.

**Data frequency**: The data frequency or sampling rate for formation control depends primarily on agent dynamics (faster moving agents needs higher sampling frequency to maintain formation accuracy) and formation precision (precise formation control requires higher data frequency). In this paper, the agents are moving at an approximately 1 m/s velocity; hence, we considered 100 Hz as the data sampling frequency to maintain the preciseness at a cm level.

**Noise reduction techniques**: Noise reduction techniques play a crucial role in improving the performance and reliability of formation control algorithms, especially in scenarios where sensor data is subject to various sources of noise and uncertainty. The Kalman filter [71] is a widely used technique for estimating the state of a dynamic system while accounting for measurement noise. In formation control, it can be employed to filter noisy sensor measurements, such as GPS positions or IMU data, to obtain more accurate estimates of agent positions and velocities. Additionally, sensor fusion (combining data from multiple sensors) and Predictive Filters (such as the Alpha–Beta filter which can provide a more stable estimate of the current state) can be employed to achieve the desired level of noise reduction and robustness in real-world formation control systems.

**Sensor synchronization**: Sensor synchronization in formation control is the process of aligning the data from sensors on different agents or vehicles in a formation such that they share a common time reference and are temporally aligned. This synchronization is crucial for achieving accurate and coordinated control of the agents within the formation. However, sensor inaccuracies can lead to errors in position estimation, which can degrade the formation quality. Additionally, delays in communication can disrupt the synchronization of sensors.

**Calibration**: In many formation control scenarios, agents may have different types of sensors, each with its own calibration requirements and limitations. Coordinating the calibration of heterogeneous sensors can be challenging, as the calibration process for one sensor may not be directly applicable to others.

**Scalability**: Scalability issues can arise when the formation control system encounters challenges in maintaining performance, coordination, and communication efficiency as the system scales up as the number of agents in the formation grows, communication load and overhead can increase significantly. Also, as the density of agents in a formation increases, the likelihood of collisions or near misses can also rise. To address scalability issues in formation control, decentralization and sparse communication can be adopted, which is scope of future exploration.

**Computational demands for real-time data acquisition and processing**: As the multi-agent system are connected with each other wirelessly, the computational demand of collecting and transmitting data wirelessly in real time to the server depends on the sampling frequency and data latency. Sometimes, the wireless transmission experience data packet losses which need to recovered, as well in the data server [3]. Efficient wireless protocols and technologies as well as a real-time operating system could be helpful in this regard. Once the data are stored in the server and sparsely populate the spatiotemporal data matrix, the real-time full-field response estimation is very fast. In this paper, each of the rows of the spatiotemporal response matrix requires approximately 0.43 s on average from the sparse data.

## 7. Discussion

Mobile sensing serves as an alternative to fixed sensing for the acquisition of vibration response data in the field of structural health monitoring. Currently, mobile sensors are operated by assigned drivers, a potentially impractical approach if the array of mobile sensors needs to follow specific patterns to optimize the data collection procedure. To address this issue, we introduce an automated multi-agent mobile sensing framework in this paper. Our proposed method diverges from fully autonomous vehicles which necessitate numerous sensors to maintain the vehicle’s position and speed, a potentially economically inefficient arrangement for structural health monitoring objectives. Therefore, the proposed formation control strategy relies solely on vehicles that autonomously manage themselves by gauging the relative velocity and relative position of their nearest neighboring agent/vehicles. In this technique, very few vehicles (often just one) with information about their global position, referred to as the “leader vehicle,” are required. This formation control strategy could be useful across various mobile sensing-oriented structural health monitoring technologies. In this work, we utitize the suggested framework for estimating full-field responses using a limited number of mobile sensors. We consider two distinct formations: Formation-1 involves two groups of vehicles crossing each other and traversing back and forth over a bridge during the data collection phase. However, Formation-1 exhibits notable response estimation errors near the bridge’s ends due to sensor gaps as the vehicle groups intersect. In contrast, Formation-2 features vehicles moving back and forth locally, resulting in a reduced number of estimation errors. Furthermore, we showcase the capability of mobile sensors to achieve any formation from any initial position and velocity using the proposed formation control framework. This strategy holds the potential to facilitate real-time assessment of changing system parameters or automated localization of damage.

## 8. Conclusions and Future Work

In this paper, we explored a minimalistic and cost-efficient scenario where the multi-agent system exclusively relies on neighboring connections. Instead, if the number of connections between the multi-agent system increases, then the consensus, as well as the whole framework, i.e., formation control combined with controlling the position of the entire formation over time, is more robust. However, achieving such a dense connection among the multi-agent systems would demand a better and larger number of sensors, potentially leading to higher costs.

This paper primarily investigates mobile sensors with time-invariant graph interaction topology. Exploring directed, weighted, and switching graph topologies could offer insights into the performance of formation control in the domain of vibration sensing and health monitoring, which remains a subject for future investigation. Additionally, in the paper, it is assumed that the vehicles are configured as point mobile sensors with unidirectional movement. In the context of 2D and 3D structures, the adaptation of mobile sensors into multi-dimensional vehicles and the potential consideration of mobile sensors as rigid bodies could be a scope of future study. Furthermore, for 2D and 3D systems, optimal paths for mobile sensors could be identified to maximize sensing information—an aspect not necessary for the current 1D movement scenario. Furthermore, to evaluate the effectiveness of the proposed method, real-life or laboratory experiments can be executed, which could also be explored in future work.

## Figures and Tables

**Figure 1 sensors-23-07848-f001:**
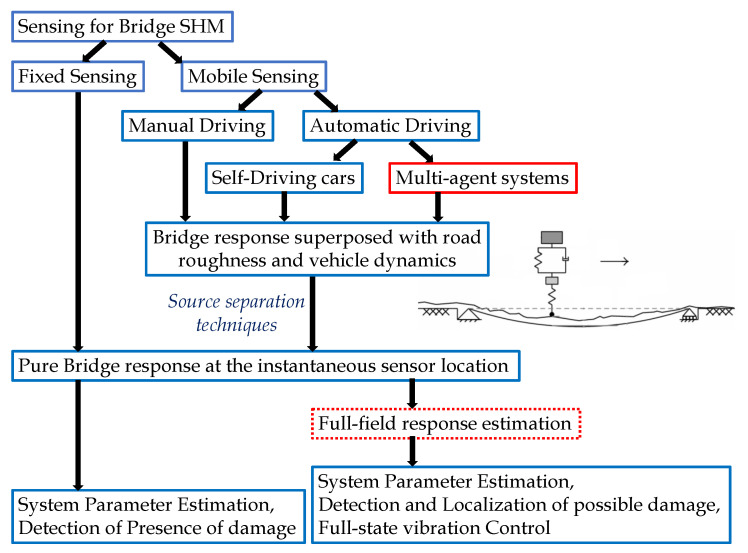
Framework for bridge vibration response sensing and structural health monitoring. This paper proposes a multi-agent formation control strategy for automatic driving in mobile sensing (marked with a red solid border). The result is numerically validated in the full-field response estimation (marked with a red dotted border). All the other framework components are marked with blue solid border. The image on the right shows that the acquired vibration response in the sensor contains the bridge vibration, road roughness, and vehicle dynamics.

**Figure 2 sensors-23-07848-f002:**
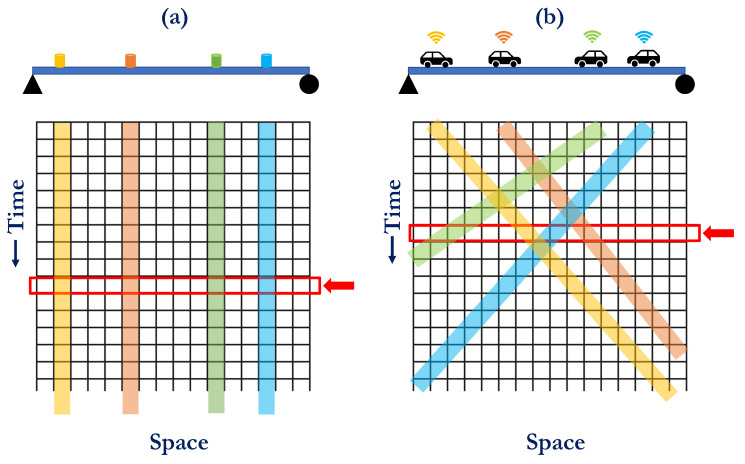
Vibration response sensing and the sensed entries of the spatiotemporal response matrix using (**a**) fixed sensors, (**b**) mobile sensors.

**Figure 3 sensors-23-07848-f003:**
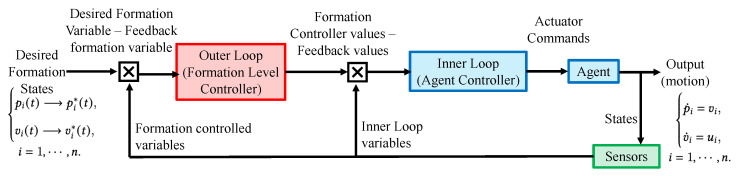
Formation control framework for a multi-agent system.

**Figure 4 sensors-23-07848-f004:**
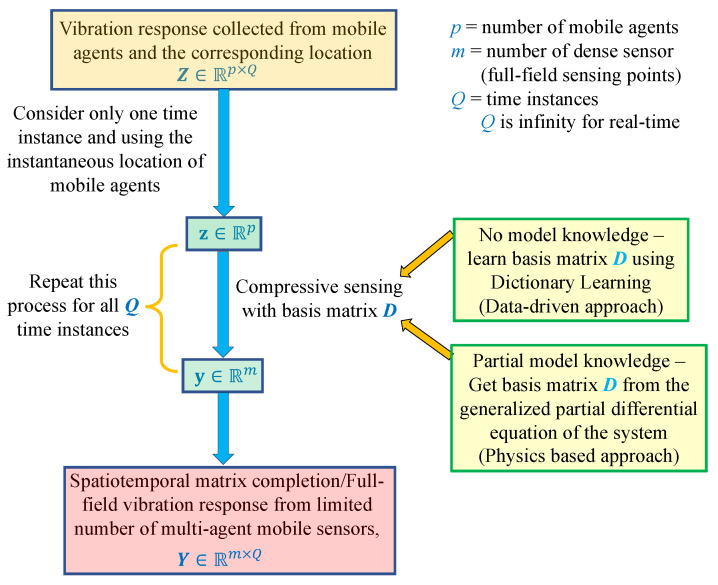
Proposed compressive sensing framework for spatiotemporal response matrix completion.

**Figure 5 sensors-23-07848-f005:**
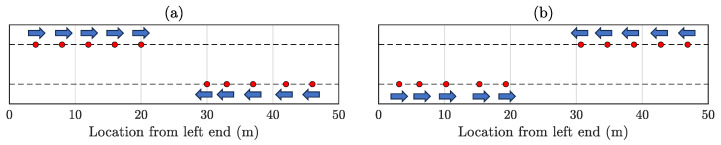
Two instances illustrating Formation 1—(**a**) R${}_{1}$ to R${}_{5}$ commencing from the left section of the bridge, while the remaining half (R${}_{6}$ to R${}_{10}$) initiate movement from the right segment. The ”blue” arrows represent the immediate direction of motion for the mobile sensors. (**b**) Once the mobile sensors detect proximity to the bridge’s end supports, they alter their movement direction. This sequence persists during the entire sensing duration.

**Figure 6 sensors-23-07848-f006:**
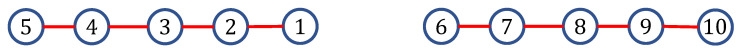
Graph connection topology of Formation-1. Mobile agents are connected only with their neighboring agents. The red line shows the connection between the agents.

**Figure 7 sensors-23-07848-f007:**
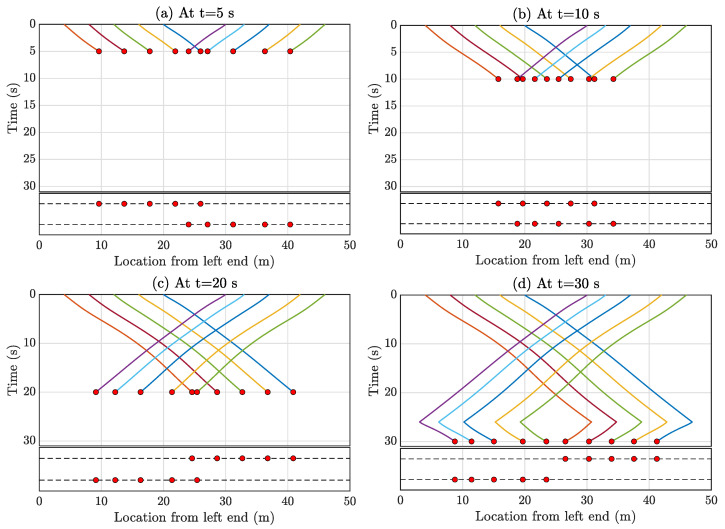
Sparsely filled entries of the spatiotemporal matrix at various time instants due to the data collected by automatic multi-agent mobile sensors in Formation-1. (**a**) at t = 5 s, (**b**) at t = 10 s, (**c**) at t = 15 s, and (**d**) at t = 20 s. The top figure of each subfigure shows the filled entries, and the corresponding bottom figure shows the instantaneous position of the mobile sensor formation.

**Figure 8 sensors-23-07848-f008:**
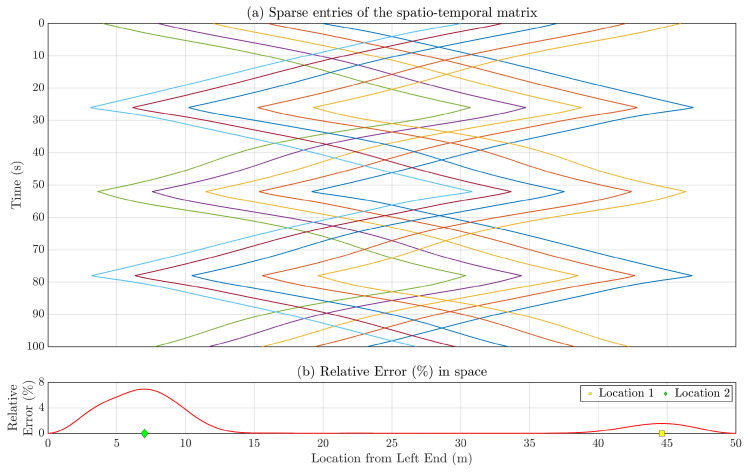
(**a**) Sparse entries in the spatiotemporal response matrix for the entire duration of 100 s using Formation-1 which are used for estimating full-field response. (**b**) Relative reconstruction error (%) is associated with each location along the length of the simply supported bridge. Location 1 corresponds to the highest error observed across the bridge’s entire span.

**Figure 9 sensors-23-07848-f009:**
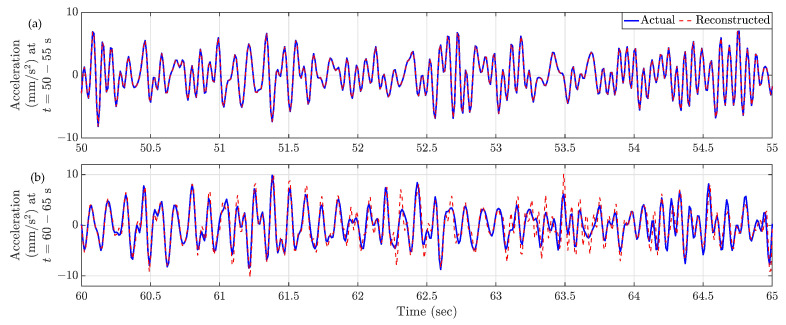
Comparison of reconstructed and actual time history responses at Location 1 (Figure 8) for two time snippets. (**a**) t = 50–55 s, (**b**) t = 60–65 s.

**Figure 10 sensors-23-07848-f010:**
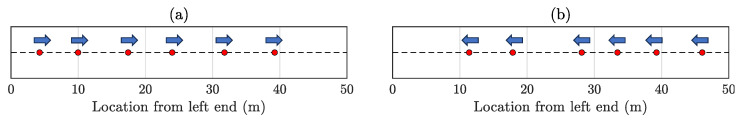
(**a**) Initial positions of R${}_{1}$ to R${}_{6}$ in Formation-2. The ”blue” arrows indicate the current movement direction of the mobile sensors. (**b**) When the mobile sensors detect their proximity to the bridge end supports, their movement direction is altered. This back-and-forth process continues throughout the data collection phase.

**Figure 11 sensors-23-07848-f011:**
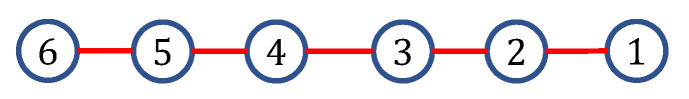
Graph connection topology of Formation-2. Mobile agents are connected only with their neighboring agents. The red line shows the connection between the agents.

**Figure 12 sensors-23-07848-f012:**
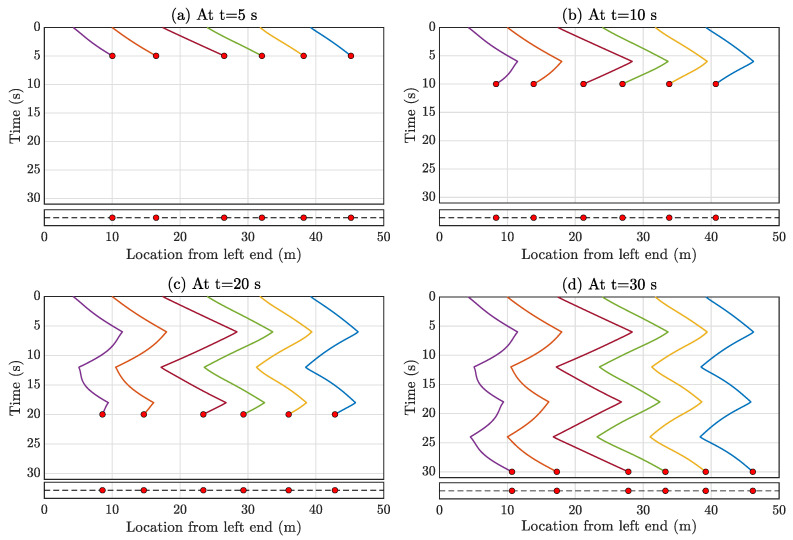
Sparsely filled entries of the spatiotemporal response matrix at different time instants due to Formation-2 motion of multi-agent mobile sensors at (**a**) at t = 5 s, (**b**) at t = 10 s, (**c**) at t = 15 s, and (**d**) at t = 20 s. The top figure of each subfigure shows the filled entries, and the corresponding bottom figure shows the instantaneous position of the mobile sensor formation.

**Figure 13 sensors-23-07848-f013:**
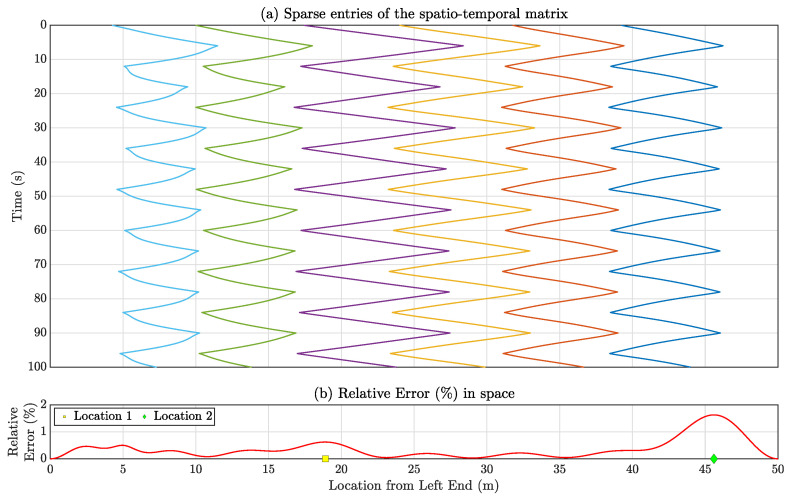
(**a**) Sparsely filled sparse entries of the spatiotemporal matrix for the total time 100 s with Formation 2. (**b**) Relative reconstruction error (%) for each location of the simply supported bridge.

**Figure 14 sensors-23-07848-f014:**
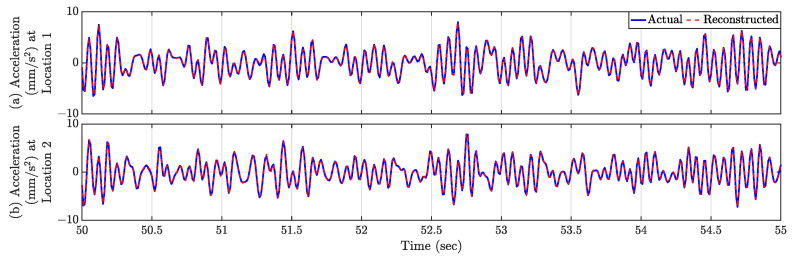
Time history response comparison of (**a**) Location 1 and (**b**) Location 2 in Figure 13b. Both the reconstructed time histories are comparable with the actual time histories.

**Figure 15 sensors-23-07848-f015:**
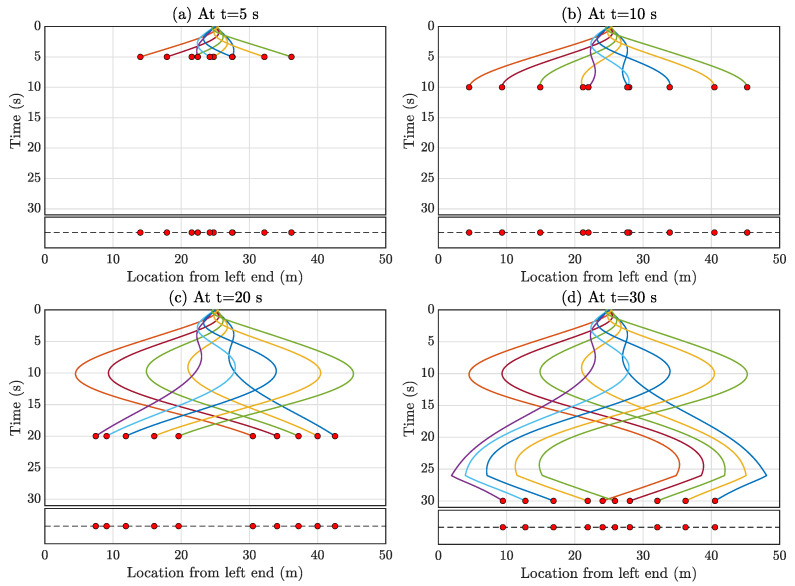
The automated multi-agent mobile sensors achieve Formation-1 where the initial position of all the mobile sensors is in the middle of the bridge. This figure shows the instances of how it achieves the target formation (**a**) t = 5 s, (**b**) t = 10 s, (**c**) t = 15 s, (**d**) t = 20 s.

## Data Availability

Data will be made available on request.

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
