# Peer review of "Full-Field Vibration Response Estimation from Sparse Multi-Agent Automatic Mobile Sensors Using Formation Control Algorithm"

_sensors, 2023, doi:10.3390/s23187848_

Round 1

Reviewer 1 Report

The author's research is of great significance for future bridge health monitoring. The article is well written. However, some small details need to be revised. Therefore, I recommend it to be accepted after a minor revision.

The specific content is as follows:

1. These title (1.1. Motivation, 1.2. Literature Review and Gaps, 1.3. Unique Contribution and  1.4. Organization) suggestions are deleted.

2. 1.4. Organization: It is recommended to rewritten it to the brief content, purpose and significance of this study. There is no need to summarize the sections.

3. 6. The conclusion part is too long. It is recommended that this part is divided into 6. Discussion and 7. Conclusion: two sections.

4. References formats are revised one by one according to the requirements of the journal.

Reviewer 2 Report

The paper

“Full-field vibration response estimation from sparse multi-agent automatic mobile sensors using formation control algorithm”

By Jana & Nagarajaiah (UCLA and Rice University)

focuses on the application of mobile sensors for structural vibration response sensing. According to the Authors, this can offer advantages in terms of flexibility and spatial information acquisition.

The paper specifically addresses the challenge of costly self-driving automated vehicles for structural health monitoring by introducing a formation control framework for automatic multi-agent mobile sensing. The proposed formation control algorithm manages the behaviour of multi-agent systems for structural response sensing using vibration data collected by mobile sensors.

The paper presents results from applying this approach to a simply supported bridge, demonstrating the capabilities of the proposed method for the reconstruction of the vibration response and highlighting the potential for automated structural response measurement for health monitoring and resilience assessment.

Overall, the concept is of interest to researchers in the field of bridge monitoring and the paper is remarkably well-made. Hence, acceptance can be granted; only a few, minor and non-binding comments are reported hereinafter,

-        Figures 7 and 12: zoom boxes could be useful to better see the small differences between the actual and reconstructed signals.

-        A compressive sensing algorithm is employed to estimate the full-field vibration response of the structure based on the sparsely populated spatiotemporal response matrix obtained from mobile sensors. The basics are recalled in Section 4.1; however, a somehow slightly more detailed discussion would be helpful for the readers.

-        The Conclusions are a bit lengthy and could be shortened.

The English is overall fine and the writing of the paper is good.

Reviewer 3 Report

a. Address the limitations of using self-driving vehicles for data collection in structural health monitoring.
b. Elaborate on how "user choice" influences the proposed formation control algorithm.
c. Provide a detailed comparison of the chosen compressive sensing approach with alternative methods.
d. Quantitative metrics for evaluating the algorithm's performance need to be defined.
e. Explain the significance of the "spatiotemporal response matrix completion task" in depth.
f. Offer a step-by-step breakdown of how the compressive sensing technique is applied to collected data.
g. Describe the experimental setup, sensor arrangement, data frequency, and noise reduction techniques.
h. Discuss limitations like sensor synchronization, calibration, and scalability for complex structures.
i. Address computational demands, especially for real-time data acquisition and processing.

I have no suggestions
